# Vegetable Cellulose Fibers in Natural Rubber Composites

**DOI:** 10.3390/polym15132914

**Published:** 2023-06-30

**Authors:** Elizabeth R. Lozada, Carlos M. Gutiérrez Aguilar, Jaime A. Jaramillo Carvalho, Juan C. Sánchez, Giovanni Barrera Torres

**Affiliations:** 1Faculty of Arts and Humanities, Metropolitan Institute of Technology-ITM, Medellín 050036, Colombia; 2Advanced Manufacturing Technology Center, SENA, Medellín 050036, Colombia

**Keywords:** vegetable fibers, natural rubber, composites, lignocellulosic fibers

## Abstract

In the last decade, natural fibers have had a significant impact on the research and development of innovative composites made with natural rubber, improving their properties over those of their counterparts that incorporate polluting synthetic fibers. In recent years, this fact has stimulated the research into several modified natural rubber composites reinforced with vegetable fibers. This paper reviews the scientific literature published in the last decade about the properties and characteristics of natural vegetable fibers and natural rubber used in composites. Nowadays the use of alternative materials has become necessary, considering that synthetic materials have caused irreversible damage to the environment, being associated with global warming, for this reason research and development with materials that print a lower carbon footprint during the manufacturing process and subsequent product manufacturing. This review is an invitation to the use of vegetable fibers, as well as vegetable-type matrices, in this case natural rubber as a binder system, it is fantastic to know the different works carried out by other scientists and engineers, in this way to project new compounds linked to innovation in processes that reduce the carbon footprint and its negative impact on our planet.

## 1. Introduction

The overexploitation of oil resources, which are estimated to last no more than 60 years at the current consumption rate [1], and the exhaustion of nonrenewable resources have encouraged the reduction of environmental impacts, increasingly raising environmental awareness in the areas of health care, sustainability, and conservation of the ecological balance. This has also caused the prioritization of biodegradable and sustainable raw materials, as well as the development of much more environmentally friendly ones [2,3,4,5] as substitutes in conventional industrial applications that present low sustainability. Raw materials such as natural rubber and vegetable fibers are considered appropriate to replace similar synthetic materials from fossil sources that are currently used to make many products.

The worldwide interest in natural fibers in different applications has increased in the last decade. Only in 2018, the global natural fiber production reached 32 million metric tons [6]. The main reason for the increase in demand for this raw material is that, thanks to its physical-mechanical properties, it can be used in multiple applications for industrial development [7].

Compared to inorganic fibers, natural fibers are abundant renewable resources. They are easily biodegradable; nontoxic; nonabrasive; low-density; low-cost [5,8,9,10,11,12,13,14]; not harmful to human health [15,16]; and resistant to corrosion [17], alkali, and most organic acids. However, they are destroyed by strong mineral acids [18]. Natural vegetable fibers (VF) have been widely used to reinforce polymeric composite materials [19,20]. Besides, VF (in percentages previously analyzed in several studies) are preferred over inorganic fibers in industrial applications and to develop new composites due to their low cost, versatility, abundance, environmental attributes (i.e., renewable, recyclable and biodegradable) [13,15,21,22], mechanical properties (e.g., high tensile strength, flexural modulus, and low density), and high electrical resistance, thermal and acoustic insulation, and fracture resistance [23]. At the same time natural fibers offer multiple benefits, studies also show that their mechanical properties are low compared with synthetic fibers due their hydrophilic nature. Therefore, they require a surface treatment to improve their adhesion and coupling to the polymeric phase, improving interfacial adhesion and its mechanical response in polymeric matrices.

## 2. Composite Materials

Composite materials are made artificially by combining two or more material families, which produces an improved material with properties derived from its constituents. Composite materials are made up of two important phases: (1) the matrix, which bonds and protects the material constituents and transfers stress to the second phase; and (2) reinforcement, which provides strength and stiffness [24]. The resulting properties of composite materials depend to a great extent on the characteristics of the matrix; the type of reinforcement; the size, shape, and dispersion of the reinforcing particles and fibers; and a good interaction between the two phases [16,25].

In the first polymeric composites reinforced with vegetable fibers (VF), common synthetic fibers, such as fiberglass, were replaced with natural fibers in polymeric matrices [26,27]. Currently, petroleum is one of the most important non-renewable natural resources, and a large number of raw materials are synthesized from it; therefore, it has had a high demand in the market, and its global reserves have been severely reduced [28]. Fossil fuels are the main cause of pollution [29] and global warming, which is why their consumption should be strictly reduced. The mass production of polymeric composites with VF started in 1908 [30,31]; however, this type of composites with polymeric binders had already been used since 1896 in airplane seats and fuel tanks [21,26,32]. Since 1990, VF have been a more eco-friendly, economical (1.2–1.6 g/cm^3^) [33,34], and healthier alternative to fiberglass (2.4 g/cm^3^) as reinforcing raw material [35] in the automotive industry, where they have been studied in depth. 

Nowadays, more environmentally friendly composites, which include natural rubber, are being developed [4,12,35,36], and the global polymer production from sustainable natural sources has grown more than 400% between 2016 and 2019 [37]. These materials, which are also known as green composites, eco-composites, bio-composites, or eco-friendly composites, have ecological and environmental properties comparable to those of conventional composites [12,27,29,38]. 

Sustainable alternatives to oil-derived polymers obtained from renewable resources can reduce the carbon footprint of the products they are part of [39]. Natural rubber is obtained from renewable sources, making it an attractive environmentally-friendly option to develop eco-friendly composites. Natural rubber composites reinforced with vegetable fibers are interesting in the industry from the points of view of engineering, economics, and the environment because they reduce the use of dangerous active fillers [5]. The main mechanical properties of natural rubber composites reinforced with natural fibers are higher strength and modulus but show decreasing elongation at failure [34,35]; in comparation to particulate-filled in natural rubber composites [40]. Composite materials made of natural rubber and short fibers have been used in the industry to develop several commercially viable products, such as conveyor belts [17], soles [34], and V-belts [41,42], optimizing their strength, stiffness, modulus, and damping [43]. This review presents a general overview of current applications, research, and development of composite materials such as natural rubber reinforced with natural vegetable fiber. Based on a compilation of documents and studies published over the last eleven years, this review provides critical elements to explore natural fibers, which can result in cleaner production with a lower environmental impact. 

## 3. Vegetable Fibers Commonly Used in Composite Systems 

More than a century ago, most raw materials used to manufacture commercial and technical products (e.g., ropes, textiles, and canvas) were sourced from natural fibers present in different regions; even today, products such as paper are still manufactured using those fibers [31]. Nowadays, VF are the most highly valued and widely studied natural fibers in the industry and science, which is mainly due to their environmental properties, availability, abundance, and quick acquisition [38]. VF are classified according to the part of the plant from which they are extracted, such as bast, leaves, seeds, fruits, wood, and reeds (straws, husks, and roots) [44], as shown in Figure 1. VFs can also be classified into primary and secondary fibers according to their utility. Primary fibers are extracted to be used exclusively as fibers (e.g., hemp, jute, sisal, and kenaf). Secondary fibers are derived from other market needs, e.g., pineapple leaves and crown, which are by-products [26,45,46]. Some of the most widely studied natural vegetable fibers are extracted from the bast, stem, or phloem (of banana, flax, hemp, jute, and kenaf); leaves (of abaca, agave, banana, pineapple, palm, and sisal); seeds and fruits (of coconut, coir, cotton, kapok, and rice husk); reeds (of bamboo, barley, corn, nettle, oats, cane, rice, rye, and wheat); and wood [8,15,18]. 

Reinforcing vegetable fibers in composite materials are not totally homogeneous due to growth conditions (temperature fluctuations, soil, water, etc.) and extraction or harvesting process; hence, these variables and their origin should be considered from the superficial level (i.e., heterogeneous defects in their length) to their growth and cutting process [1,48,49]. Several countries worldwide produce outstanding amounts of VF, as shown in Table 1, except for wood fibers and cotton. Bast fibers are the most widely used lignocellulosic fibers due to their superior technical characteristics and easy and fast production and extraction [6,38,50].

Natural fibers represent an important socio-economic factor worldwide. They are grown in rotation with food crops and commercialized in at least 100 countries, generating a productive value of 60,000 million dollars. If natural fibers are packaged under international standards, they present no loss in value due to storage longevity, quality reduction, damage, biological attack, humidity, or the environment [6]. Asian countries produce the largest amounts and widest variety of natural vegetable fibers. In South and North America, Brazil, Colombia, Venezuela, the US, and Canada are the biggest producers of natural fibers. Some of these countries share a warm climate.

Knowing the chemical composition of composites reinforced with natural fibers is key to understanding the influence of their characteristics and determining an adequate production process for them [8]. Vegetable fibers are also called lignocellulosic fibers because they are made of hemicellulose, lignin, and cellulose, three fundamental biopolymers that constitute and shape plants. Waxes, pectins, and water-soluble substances are also found in vegetable fibers. The percentage of these components depends on the type of fiber (origin); likewise, their quantity determines the physical and mechanical properties of natural fibers [19]. Table 2 shows the physical and mechanical properties of some vegetable fibers. 

The chemical and structural composition, microfibrillar angle, and imperfections of fibers define their general properties. Therefore, the quantity of the components in the fibers should be determined because it changes depending on their species, age, or part of the plant from which they are extracted. The composition and properties of lignocellulosic fibers are difficult to establish because of the heterogeneous distribution of their basic components along the length of the plant. Table 3 details the chemical composition of some of the most commonly studied fibers.

Once the part of the plant from which the fibers are obtained is collected, the next step is retting. In the latter, the components that are not required (i.e., hemicellulose, lignin, waxes, pectins, etc.) are removed, leaving mainly cellulose [15], which is the strongest and most rigid component in the fiber.

Cellulose is the most abundant biopolymer on the planet [61]. It has been estimated that plants produce 1 trillion (10^12^) tons of cellulose annually by photosynthesis [15]. Cellulose is also the most important structural component in most natural vegetable fibers since it provides strength and constitutes up to 70% of said fibers [8,12]. The cellulose contains three hydroxyl groups (–OH) that form intramolecular and intermolecular hydrogen bonds [23,25] consists of carbon (C), hydrogen (H), and oxygen (O) with the formula C_6_H_10_O_5_. In terms of abundance in a cellulose macromolecule, lignin and hemicellulose are in second and third place, respectively [47]. Lately, the development and applications of cellulose nanofibers (CNFs) have progressed dramatically [62], for example, to reduce the use of CB in rubber composites. Some studies have tried to use some non-black fillers, especially silica in order to replace CB in rubber formulations as a substitute for carbon black in automobile tires with success on these applications [63].

## 4. Natural Rubber

Cis-1,4-polyisoprene, the main component of natural rubber and an elastomer polymer, has been used worldwide for a long time because of its low price and non-toxic characteristics. Natural rubber is obtained from latex (i.e., a milky and thick fluid), which is processed sap produced by Moraceae and Euphorbiaceae. However, the primary source for the commercial extraction of natural rubber is the tree *Hevea brasiliensis* [64,65,66]. There is significant industrial interest in natural rubber because it exhibits remarkable flexibility and impermeability. Natural rubber is a basic biomaterial that cannot be replaced with its synthetic equivalent in some industrial applications since the latter exhibits poor properties [64,65]. Due to its molecular structure and high molecular weight (>1 MDa), natural rubber is elastic, impact and abrasion resistant, efficient heat dispersal, and resilient and malleable at low temperatures [5]. All these properties and characteristics constitute an attractive alternative to the polymers traditionally used in polymeric composites with natural fibers [42].

Natural rubber is used to manufacture many essential products, such as surgical gloves, hoses, condoms, elastic bands, baby bottle nipples, and adhesives, but its main application is in the automotive industry, where 67% of the worldwide production of this raw material is used to produce tires and inner tubes [64]. Natural rubber is essential in engineering applications where it cannot be replaced with synthetic rubber [67]; for this reason, its use has increased in some industrial sectors such as the automotive industry [45,68,69,70]. Currently, Asia is the biggest producer of rubber, with 94% of the world production, while South America produces only 2% [66]. Today, the market offers several types of rubber obtained from latex coagulation. The types of rubber most commonly used in the industry worldwide are technically specified rubber (TSR), standard Malaysian rubber (SMR), dark Brazilian granulated rubber (GEB-1), Brazilian pale crepe (CCB-1), and rubber smoked sheet (RSS-3). These five types of rubber are implemented in the industry because they are available and economically viable. The acronyms stand for the standard names established by the country where the rubber is produced, and the number indicates the quantity of impurities in hundredths of a percent.

Over the last 11 years, several studies have examined the viability of multiple natural vegetable fibers as alternatives to conventional ones, e.g., rattan (*Calamus manan*); areca (*Areca catechu* L.) [71]; hibiscus (*Hibiscus sabdariffa*) [72]; nettle [73]; wheat, rye, and oat stalks [26,40]; sabra; and alfa [57,74,75]. In addition to the morphological properties and features of these fibers, their physical-mechanical characterization in natural rubber composites has also been investigated. Several factors influence the use of natural vegetable fibers in scientific studies, such as their origin, the available amount of this resource, and the number of hectares planted. Between 2016 and 2018, some of the natural fibers most commonly investigated as reinforcing materials in composites were cotton, pineapple fiber, bamboo, and flax [38,76], as shown in Figure 2.

The research and development of composite materials made with natural rubber and vegetable fibers have presented a growing trend in the last decade and the first half of 2021. This fact is confirmed by the bibliometric analysis in Figure 3. The latter shows the number of research articles in the Scopus database that were retrieved using the following keywords: “natural rubber,” “natural fiber,” “composite”, “lignocellulosic,” “plant fiber,” “plant-based natural fiber,” “bio-composite,” “natural fibers,” and “vegetable fiber.” These keywords were combined with Boolean operators (e.g., OR and AND) and quotation marks (“”), and the results were limited to document published between 2010 and 2021.

Using the search criteria and Boolean operators mentioned above, 85 documents about this topic published until June 2021 were retrieved from said database. According to Figure 3, a significant number of studies on composite materials made of natural rubber and vegetable fibers were published in 2013 and 2016, as well as in the last decade. Figure 4 shows the countries that have published most studies on these composite materials.

Among the 25 countries in Figure 4, India, Malaysia, and Thailand (Asian countries) present the widest dissemination of and most research into composites made with these renewable raw materials. These results can be explained by the fact that Asia is the biggest producer of rubber worldwide, as well as the biggest producer and consumer of vegetable fibers around the world [6,67,76].

## 5. Natural Rubber Composites Reinforced with Vegetable Fibers

After reinforcing materials, fillers are the second most important component (per volume) in rubber composites. They improve the process (to optimize extrusion capacity and reduce energy consumption) and aesthetics (texture and color) and act as diluents (to reduce the cost of the final product) or as reinforcing agents (to improve mechanical properties such as hardness, tear strength, and toughness), which is their most important in industrial application [77]. Natural rubber composites reinforced with different fibers have been extensively studied by many authors [40]. Particles sourced from natural systems can improve the physical and mechanical properties of polymer matrix composites [10], transforming them into materials with excellent mechanical properties. Thanks to their properties and characteristics, natural vegetable fibers are suitable substitutes for their synthetic counterparts, which are used to make many of today’s products.

Although natural rubber composites reinforced with vegetable fibers provide good strength and dimensional stability, their mechanical properties usually depend on how the matrix transfers stress to the fiber. The most important properties of fibers are strength, aspect ratio, length [38,78], rubber/fiber interaction (i.e., the compatibility of these phases), and interfacial adhesion [5,17,42]. Table 4 details the natural vegetable fibers most widely studied in recent years (i.e., water hyacinth, sabra, alfa, rye, oat, *Cuscuta reflexa*, and rattan particles), in addition to fibers that have been traditionally investigated (i.e., jute, pineapple leaves, sugar cane bagasse, sisal, and hemp).

Between 2010 and part of 2021, jute fibers (in natural rubber and vegetable fiber composites) were the most widely researched material. Jute is an important agricultural product for the Asian economy and one of the most common fibers in countries such as India, China, and Bangladesh. It is also produced in some Latin American countries and exhibits excellent mechanical properties compared to other fibers [52,103]. The second most commonly studied fibers are cereal straws (oats, rye, wheat, triticale, and rice husks) from agro-industrial waste generated in the production of several consumer cereals. Like jute, these fibers are mainly produced in China and India. The latter, in addition to Africa, the US, and some Latin American countries, are the biggest producers of these grains around the globe [103,104].

Another important fiber, *Cuscuta reflexa* (also called dodder plant) is native of Asia, and it is considered a highly prolific parasite and a pest in Southeast Asian countries. It has also been studied to treat multiple human health conditions [88].

### 5.1. Surface Modification of Vegetable Fibers (Improving the Characteristics of Natural Fibers in Natural Rubber Composites)

Lignocellulosic fibers present technical barriers to manufacture composites due to their variation in cell wall structure, composition, and geometry [15,105]. The main disadvantage of natural fibers as reinforcing components in composite materials, compared to synthetic fibers, is their hydrophilic nature, which makes them polar and therefore not compatible with the non-polar hydrophobic polymeric matrix [8,13,15,32]. Lignocellulosic elements, pectins, waxes, and natural plant oils act as barriers of the reactive groups of these fibers in the matrix.

The interfacial interaction between fiber and matrix is considered a critical factor in the production of high-performance composites and their applications [15,17,61,96]. Weak interfacial adhesion implies incompatibility between composite phases, leading to poor mechanical properties [4,14,106,107]. The compatibility of the fiber surface with the matrix phase depends to a large extent on the appearance of the fiber; the higher the roughness, the more anchorage points of the coupling and reinforcement material [107], improving the permeability and dispersion of the fibers. However, poor wetting of the fibers results in a non-uniform distribution of said fibers in the matrix and the formation of voids [31]. Therefore, the fibers should be subjected to physical or chemical treatments [13] that break down the hemicellulose and lignin surrounding the cellulose in order to facilitate the interaction between the fibers and the matrix, generating greater contact between them. Although physical treatments improve the characteristics of the composites reinforced with lignocellulosic fibers, they entail a higher cost than their chemical counterparts, which are the most widespread option for surface modification of natural fibers [21]. Chemical treatments reduce the -OH functional groups present on the surface of the natural fiber and increase its roughness [33].

Several types of chemical treatments can be applied to fibers to modify their morphology: silane, alkali, acrylation, benzoylation, maleated coupling agents, permanganate, acrylonitrile grafting and acetylation, stearic acid, peroxide, isocyanate, triazine, fatty acid derivative (oleyl chloride), sodium chloride, and fungi [58]. Several authors have attempted to improve the compatibility between natural fibers and rubber by modifying the fiber surface [44,55,74,75,96].

Undoubtedly, the most common chemical process for fibers is the alkaline treatment [21], which uses sodium hydroxide (NaOH) as alkali [107]. In this treatment, also called mercerization [108], the fibers are immersed in NaOH solutions at specific concentrations to remove lignin and surface impurities (such as pectins and oils), which act as barriers and reduce the hydrophilic behavior of the hydroxyl groups in the fibers. As a result, the roughness of the fiber surface is increased [109]. Fibers pretreated with alkaline agents, even in low proportions, have been shown to favor the mechanical properties of composites, providing higher strength, stiffness, and better interfacial ratio thanks to a higher cellulose exposure and higher surface energy, which produce better wetting and compatibility [15].

### 5.2. Applications

Composites of natural rubber filled with lignocellulosic fibers [110,111] offer many benefits because both materials come from low-cost, low-density renewable sources, and they have promising technical and engineering applications [112,113]. Several products, such as belts, hoses, and tires, are made of natural rubber and short fibers (Figure 5).

The combination of natural rubber and short lignocellulosic fibers brings about a synergistic effect, resulting in enhanced mechanical properties and improved performance of these critical components. The incorporation of the fibers reinforces the natural rubber matrix, leading to increased strength, durability, and resistance to wear and tear [83]. A significant advantage of composites of natural rubber filled with lignocellulosic fibers is their biodegradability. As both natural rubber and lignocellulosic fibers are biodegradable materials, they offer a sustainable and environmentally friendly alternative to traditional synthetic materials. This eco-friendly nature opens up a plethora of potential applications in various fields [114], they can be used in biomedicine, agriculture, storage or packaging, and hygiene devices [16]. However, the automotive sector is the industry that has used natural rubber composites the most because of their benefits in terms of lightness and economical production of various parts. For example, Mercedes-Benz has used natural rubber/coconut fiber composites to manufacture the support of the dashboard and the surface of the seats and backrests of the C-, S-, E-, and A-class vehicles of the Mercedes 164/251 2004 model [70,115].

Natural rubber composites reinforced with sugarcane bagasse have good mechanical properties, such as hardness and Young’s modulus, but exhibit poor abrasion and stress resistance [85,116,117]. In other side, the composite materials have been implemented in footwear to prevent slipping on ice, using bamboo fibers as reinforcement to increase the Young’s modulus and hardness, which improves the coefficient of friction (COF). Although the COF decreases when the sole absorbs water, the other mechanical properties are preserved thanks to the fiber treatment [116,118]. The utilization of bamboo fibers as reinforcement in natural rubber composites presents an innovative approach to improve the coefficient of friction and address slipping concerns on icy surfaces. These advancements contribute to the development of safer and more reliable footwear solutions, particularly in challenging environmental conditions.

Further research and development efforts are being undertaken to explore novel methods for enhancing the overall performance of natural rubber composites reinforced with lignocellulosic fibers in footwear applications. The aim is to achieve superior abrasion resistance, stress resistance, and wet traction while maintaining the desired mechanical properties and ensuring long-lasting durability.

Within the construction industry, natural rubber filled with lignocellulosic fibers composites find application in a multitude of areas including building materials, roofing sheets, insulation panels, and flooring. By virtue of their enhanced mechanical properties, thermal insulation capabilities, and heightened durability, they prove to be suitable for both residential and commercial construction projects [119].

The versatility of composites of natural rubber filled with lignocellulosic fibers is continually being explored, with ongoing research and development efforts seeking to unlock their full potential. As scientists and engineers delve deeper into understanding their properties and processing techniques, new and innovative applications across various sectors continue to emerge. These composites hold tremendous promise for creating a more sustainable and environmentally conscious future.

## 6. Conclusions

Vegetable fibers used as reinforcements in natural rubber composites have great potential in engineering applications due to their technical competitiveness and physical-mechanical properties and environmental impact, which could reduce their carbon footprint compared to synthetical fibers used in conventional processes.

The results of this literature review show that the countries that have most extensively studied composite materials made of natural rubber reinforced with vegetable fibers are also the biggest producers of such fibers.

Composite materials made of natural rubber and vegetable fibers should be further researched to find ideal formulations because many variables determine their possible innovative, sustainable, and economical industrial applications.

This study discussed possible applications of natural fibers in the manufacture of natural rubber composites.

## Figures and Tables

**Figure 1 polymers-15-02914-f001:**
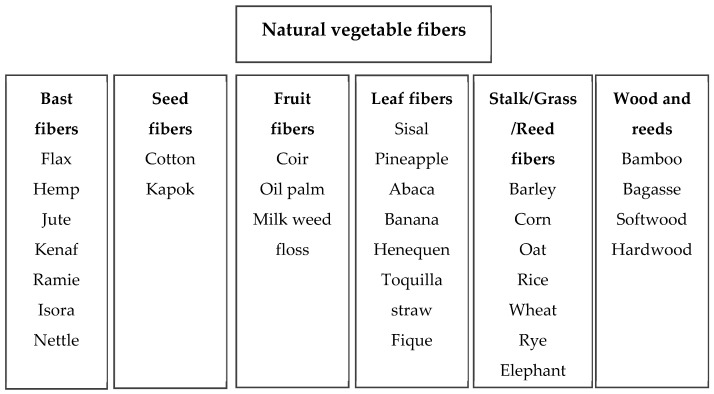
Classification of vegetable fibers according to their origin [1,4,13,15,18,24,28,36,45,47].

**Figure 2 polymers-15-02914-f002:**
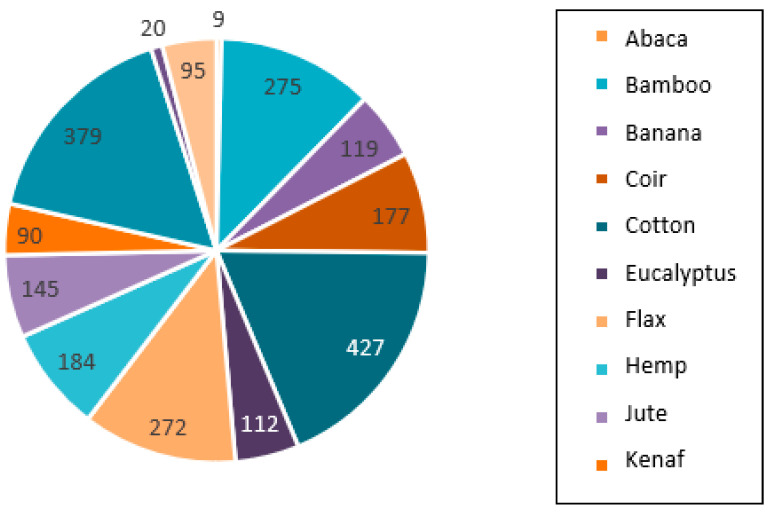
Natural fibers in composites studied between 2016 and 2018. Adapted from [38].

**Figure 3 polymers-15-02914-f003:**
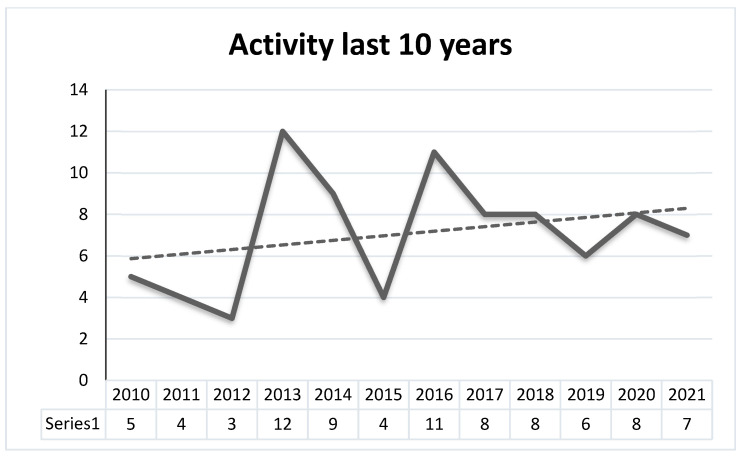
Number of documents about natural rubber composites reinforced with vegetable fibers published annually between 2010 and 2021.

**Figure 4 polymers-15-02914-f004:**
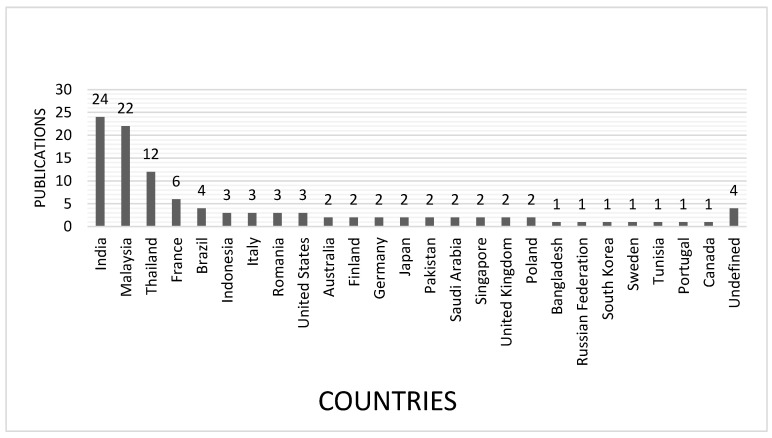
Number of documents on vegetable fiber-reinforced natural rubber composites published between 2010 and June 2021 classified by country.

**Figure 5 polymers-15-02914-f005:**
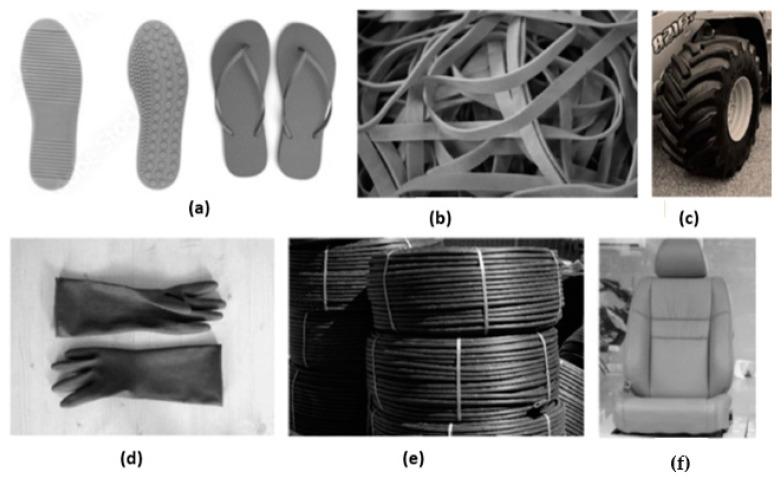
Pictures of commercial products made of composites of natural rubber and vegetable fibers: (**a**) soles and flip-flops, (**b**) belts and straps, (**c**) tires, (**d**) gloves, (**e**) hoses, (**f**) seat and backrest surface of Mercedes-Benz vehicles.

**Table 1 polymers-15-02914-t001:** Worldwide production of the most common vegetable fibers and their biggest producers [4,26,27,31,45,51].

Fiber	Main Producing Countries	World Production (10³ tons)
Bagasse	Brazil, India, China	75,000
Bamboo	India, China, Indonesia	30,000
Cotton	China, India, US	25,000
Jute	India, China, Bangladesh	2300
Kenaf	India, Bangladesh, US	970
Flax	Canada, France, Belgium	830
Sisal	Tanzania, Brazil	378
Hemp	China, France, Philippines	214
Coir	India, Sri Lanka	100
Ramie	China, Brazil, Philippines, India	100
Kapok	Vietnam, Philippines	96
Pineapple	Philippines, Thailand, Indonesia	74
Abaca	Philippines, Ecuador, Costa Rica	70
Oil palm	Malaysia, Indonesia	40
Banana	India, China, Ecuador	36
Fique	Colombia	20
Curaua (pineapple)	Brazil, Venezuela	>1

**Table 2 polymers-15-02914-t002:** Botanical name and physical and mechanical properties of some vegetable fibers. Adapted from [1,11,19,22,26,30,32,38,43,48,52,53,54,55,56,57,58].

		Physical Properties		Mechanical Properties		
Fiber	Species	Density (g/cm³)	Diameter (µm)	Tensile Strength (MPa)	Young’s Modulus (GPa)	Elongation (%)
Abaca	*Musa textilis*	1.5	10–13/150–180	430–813	12–33.6	2.9–10
Cotton	*Gossypium*	1.21–1.6	10–40	287–800	5.5–12.6	7–8
Areca	*Areca catechu* L.	0.7–0.8	396–476	147–322	1.12–3.15	10.2–13.15
Rice husk	*Oryza sativa*	0.5–0.7	–	–	–	–
Rice (stalk)		–	–	74.6	3.3	–
Banana	*Musa indica*	0.5–1.5	13–16/95–245	355–789	4–33.8	2–53
Bamboo	*Bambusoideae*	0.9	14/240–330	440	36	1.5
Coir	*Cocos nucifera*	1.19	149.5–249.8	131–174.8	4–6	29.85
Bagasse (sugar cane)	*Gramineae saccharum*	1.1–1.6	10–34/200–400	170–350	17–27	1.1–7.9
Hemp	*Cannabis sativa*	1.35–1.469	25–600	550–900	69–70	1.6–4.1
Curaua	*Ananas acutifolius*	1.4	7–10	87–1150	11.8–96	1.3–4.9
Fique	*Furcraea macrophylla*	–	–	511–635	9.4–22	–
Henequen	*Agave fourcroydes*	1.2–1.4	160–180	430–580	10–20	3–5.9
Isora	*Helicteres isora*	1.35	10.1	500–600	18–20	5–6
Flax		1.38–1.4	25	800–1500	60–80	1.2–1.6
Kapok	*Ceiba pentranda*	1.2–1.6	8–36	44.9–90	1.72–5	1.8–4.3
Kenaf	*Hibiscus cannabinus*	1.25–1.40	12–50	284–930	21–60	1.6–6.9
Nettle	*Urtica dioica*	1.51–1.6	25–40	650	38	1.7
Oil palm	*Elaeis guineensis*	0.7–1.55	150–500	248	3.2	14–25
Pineapple	*Ananas comosus*	1.44–1.52	50–300	169.5–1672	60–82	1–3
Ramie	*Boehmeria nivea*	1.3–1.55	20–280	400–938	61.4–128	1.2–3.8
Sisal	*Agave sisalana*	1.2–1.45	50–200	390–700	9.4–41	2.3–7
Jute	*Corchorus*	1.23–1.49	25–250	393–800	0.13–26.5	1.16–1.80
Hardwood	-	0.3–0.88	–	51–121	5.2–15.6	–
Softwood	-	0.30–0.59	15–80	45.5–111	3.6–14.3	4.4

**Table 3 polymers-15-02914-t003:** Chemical composition in percentages of some vegetable fibers [1,22,26,55,57,59,60].

Fiber	Cellulose (%)	Hemicellulose (%)	Lignin (%)	Wax (%)	Pectin (%)	Ash (%)
Abaca	56–63	15–25	7–13	0.1–3	0.3–1	1–3.2
Cotton	82.7–90	1–5.7	0.75–28.2	0.6	6	0.8–2
Areca	57.35–58.21	13–15.42	23.17–24.16	0.12	–	–
Rice husk	28–36	23–28	12–14	14–20	–	–
Rice (stalk)	28–48	23–28	12–16	–	–	15–20
Oat (stalk)	31–48	27–38	16–19	–	–	6–8
Banana	48–65	10.2–21	5–21.6	3–5	0.8–4.1	2.1
Bamboo	26–73.8	12.5–73.3	10.2–31	–	10	1.7–5
Coir	19.9–36.7	0.15–15.4	32.7–53.5	–	4–7	–
Bagasse (sugar cane)	32–55.2	16.8–32	19–25.3	–	8.8–10	1.5–5
Hemp	55–81	12–22.4	2.6–13	0.8	0.9	0.5–8
Barley	31–45	27–38	14–19	2–7	–	–
Curaua	73.6	5–9.9	7.5	–	–	–
Henequen	58–77.6	27–38	7–13.1	0.5	–	–
Isora	74.8	–	23	1.1	–	0.9
Kenaf	35–72	20.3–21.5	9–19	–	2	2–5.1
Kapok	35–64	22–45	15–21.5	3–5.31	23	0.5–0.8
Flax	71–72.5	18.6–20.6	2.2–2.5	1.5	0.9	13.1
Corn	38–40	28	7–21	3.6–7	–	–
Oil palm	42.7–65	17.1–35	13.2–25.3	0.6	–	1.3–6
Pineapple	57.5–82	80.7	4.4–12.7	3.3	1.1–4	0.9–4.7
Ramie	68.6–91	5–16.7	0.6–0.8	0.3	2	5
Sisal	60–78	10–14.2	8–11	2	1.2	0.6–4.2
Wheat (stalk)	29–51	26–32	17–32	6.8	–	4.5–9
Jute	45–71.5	12–21	12–13	0.5	0.2–11.8	0.5–8
Hardwood	38–50	19–26	20–30	–	0.1	0.1
Softwood	40–45	7–14	26–36	–	0.1	0.1

**Table 4 polymers-15-02914-t004:** Some natural rubber/vegetable fiber composites studied between 2010 and 2021.

Vegetable Fiber	Year	Reference
Areca nut fibers (coir/outer shell of the nut)	2015	[79]
Rattan (*Calamus manan*)	2012	[72]
Sisal	2019	[80]
Coconut particles	2019	[80]
Water hyacinth	2011, 2016	[81,82]
Cotton	2011	[2]
Sabra	2019	[74]
Hemp	2015, 2019	[83,84]
Sugarcane bagasse	2014, 2019	[10,85,86]
Cereal straw (rye, oats, wheat, and triticale)	2018, 2019, 2020	[26,40,44,87]
*Cuscuta reflexa*	2014, 2020	[75,88]
Rice husk	2012, 2018, 2020	[33,34,89,90]
Flax	2017	[5]
Kenaf	2011, 2013	[91,92,93,94]
Pineapple leaves	2017, 2020	[78,95]
Jute	2011, 2014, 2017, 2018, 2019, 2020	[2,42,73,96,97,98,99,100,101]
Banana	2013	[102]

## Data Availability

Not applicable.

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
