# Peer review of "Vegetable Cellulose Fibers in Natural Rubber Composites"

_polymers, 2023, doi:10.3390/polym15132914_

Round 1

Reviewer 1 Report

The manuscript is highly plagiarized (43%). Hence it is advised to modify the manuscript to reduce the plagiarism. Also, make the corrections according to the attached word file. Make all the grammatical corrections.

Needs corrections to the sentences bothe grammatical and structural.

Reviewer 2 Report

1. Title: This paper discussed only in the field of vegetable fibers. So the title should be revised "vegetable cellulose fibers in natural rubber composites".

2. This review is mainly carried out on papers published after 2010. However, active researches have been done by Thai researchers in the biggest NR producing country, such as N. Lopattananon, et. al, J. App. Poly. Sci. 102(2), p1974 (2006).  In this reference there are few papers from Thailand. I would like to recommend to be picked up some excellent papers related to natural fibers before 2010.

3. Lately the developments and applications of cellulose nano-fibers (CNF) have been drastically progressed, for example, CNF as a substitute of carbon black for automobile tires. In this review the present and future of CNF should be briefly discussed.   Saito.T, et al, J. Wood Sci. 56(3), p227 (2010),  Abraham. E, et al, Cellulose, 20, p1417 (2013)

not particular

Reviewer 3 Report

This is an interesting review article on Cellulosic fibers in natural rubber matrix composites, which should be useful to the readers of polymers journal. It’s technical quality and structure are satisfactory, therefore I would suggest publication after the following comments are addressed.

1. in lines 50-53, it seems a contradiction to say that the VFs have high mechanical properties but then they are inadequate as they are worse than those of synthetic fibers. Do they meet application requirements or not? The requirement for surface treatment seems to be an important drawback – could the authors commend.

2. it would be beneficial and more useful to the reader the references to be added at their specific places. e.g. line 96: ‘’Composite materials made of natural rubber and short fibers have been used in the industry to develop several commercially viable products, such as conveyor belts, soles, and V-belts [17], [33], [39], [40]. Is it conveyor belts [17], soles [33] and so on? (Similarly for the rest of the manuscript).

3. Same as per comments 1-2, in lines 92-96, The main mechanical properties of natural rubber composites reinforced with natural fibers are higher strength and modulus;… is this over particulate filled natural or synthetic rubber? Please be more precise and add reference.

4. lines 125-128 are confusing. ‘’Reinforcing vegetable fibers in composite materials are not totally homogeneous due to growth conditions…’’

5. Although mechanical properties of the fibers is given in Table 2. There is no comparison of mechanical properties with composites made of synthetic, glass or carbon fibers. Why is that? For sure every publication has many specific parameters (e.g. molecular weight of particular rubber) so a comparison is difficult, if possible at all, but perhaps if we have 10-20 works on each; synthetic, glass and natural fibers, it would be visible how these composites performance compare?

6. I suggest the authors to be careful with their expressions and read the manuscript throughout to make corrections where necessary. E.g. line 267: The most important properties of fibers are strength, dispersion, aspect ratio, content….’’ is dispersion a property? E.g. line 359 which are the ‘’environmental properties’’?

7. Please correct the abbreviations in the manuscript. e.g. abbreviation vegetable fibers (VF) is set in lines 46 and 67 but full name is used many times before and after that. Set the abbreviation at the first time used and use VF throughout the rest of the manuscript.

8. The Applications section, although useful, is very brief and needs to expand.

Minor editing of English language required

Round 2

Reviewer 3 Report

The authors have addressed my comments at a good level, therefore I suggest publication of this manuscript.

Minor things such as typos could be corrected at the proof phase.